# Design of an Automatically Controlled Multi-Axis Stretching Device for Mechanical Evaluations of the Anterior Eye Segment

**DOI:** 10.3390/bioengineering10020142

**Published:** 2023-01-20

**Authors:** Kehao Wang, Ziyan Qiu, Yiping Xie, Shuo Cai, Yang Zhao, Barbara K. Pierscionek, Jiangzhen Guo, Yubo Fan

**Affiliations:** 1Key Laboratory for Biomechanics and Mechanobiology of Ministry for Education, Beijing Advanced Innovation Center for Biomedical Engineering, Beihang University, Beijing 100191, China; 2School of Engineering Medicine, Beihang University, Beijing 100191, China; 3School of Biological Science and Medical Engineering, Beihang University, Beijing 100191, China; 4School of Artificial Intelligence, University of Chinese Academy of Sciences, Beijing 100043, China; 5Institute of Automation, Chinese Academy of Sciences, Beijing 100190, China; 6Faculty of Health, Education, Medicine and Social Care, Medical Technology Research Centre, Anglia Ruskin University, Bishops Hall Lane, Chelmsford, CM1 1SQ, UK

**Keywords:** biomechanics, stretching device, multi-axis loading, lens, presbyopia, cataract

## Abstract

The young eye has an accommodative ability involving lens shape changes to focus over different distances. This function gradually decreases with age, resulting in presbyopia. Greater insights into the mechanical properties of anterior eye structures can improve understanding of the causes of presbyopia. The present study aims to develop a multi-axis stretching device for evaluating the mechanical properties of the intact eye lens. A stretching device integrating the mechanical stretcher, motor, torque sensor and data transmission mechanism was designed and developed by 3D printing. The mechanical stretcher can convert rotation into radial movement, both at constant speeds, according to the spiral of Archimedes. The loading unit equipped with eight jaws can hold the eye sample tightly. The developed device was validated with a spring of known constant and was further tested with anterior porcine eye segments. The validation experiment using the spring resulted in stiffness values close to the theoretical spring constant. Findings from measurements with porcine eye samples indicated that the measured forces are within the ranges reported in the literature. The developed multi-axis stretching device has good repeatability during experiments with similar settings and can be reliably used for mechanical evaluations of the intact eye lens.

## 1. Introduction

The eye is responsible for perceiving the world via the processing of light that brings information about objects in the environment. This light which is reflected from objects is refracted by the two refractive components of the eye, the cornea and the lens, to reach the retina. From there the image-bearing signals are converted into electro-chemical signals and processed by the higher visual pathways. The cornea contributes two-thirds of the total ocular refractive power of the eye, whilst the lens contributes the remaining third [1]. The lens also has a crucial role in allowing the eye to focus over different distances, and this is accomplished through a shape-changing process, which is termed accommodation [2,3].

The lens has a bi-convex ellipsoidal shape and is held in place by a ring of suspension ligaments collectively called the zonule [4,5,6]. The zonule connects the lens equator to the ciliary muscle, which via contraction and relaxation transmits forces to the lens to alter its shape, thereby changing its refractive power and that of the eye. The extent to which the lens can alter its shape is directly related to its mechanical properties [2,7]. A number of studies have evaluated the stiffness of the eye lens by treating it as an elastic body [8,9,10,11]. Studies employing dynamic mechanical measuring techniques have shown that the eye lens also demonstrated viscoelastic behaviour [12,13] which may also influence the dynamic accommodative behaviour, suggesting that the time used for changing focus from far-to-near objects may not be the same as the time used for focus changes from near-to-far objects [14,15].

With age, the eye gradually loses its ability to alter its focus, and by the age of around sixty years, focus on near objects is no longer possible; this condition is called presbyopia. The onset of presbyopia can be attributed to several factors [2], including the increase in stiffness of the lens and changes in biomechanics of the lens capsule (the semi-elastic membrane which envelops the lens) [16,17,18,19], the growth of the lens that alters its size and curvature [20,21,22] and/or the decrease in contractability of the ciliary muscle [23,24]. The stiffness of the human eye lens was found to increase with age [8,9,11], which makes it harder to alter the lens shape under similar levels of stretching forces applied by the zonules, and this may partially explain the age-related decline in accommodative amplitude [2]. The age-related stiffening of the lens is considered to be the most likely cause of presbyopia. To definitively determine that this is indeed the major cause of presbyopia and to know how the mechanical behaviour of the accommodative system changes with age, a reliable and robust measuring method is needed.

Various experimental methods have been used to evaluate the mechanical properties of the eye lens. These include the seminal work by Fisher [25], who developed the spinning lens method to mimic the lens shape change during accommodation using centrifugal forces. A similar technique was investigated by Burd et al. [18], the classical compressional method adopted to characterise the overall lens stiffness [26], stretching of the lens to simulate ciliary muscle forces [27] as well as the indentation [28] and oscillational shear rheometry used on sectioned lens samples [12,13,29]. However, the results revealed by studies adopting the above methods have varied over several orders of magnitude in elastic moduli and inconsistencies in the ageing trends have also been reported [2]. More recent non-invasive methods including bubble-based acoustic radiation [30,31] and Brillouin scattering analysis [32,33] have the potential to be applied in in vivo human eyes, but these two methods report bulk modulus, which is hard to convert to Young’s or shear modulus for soft tissue [34,35], making it difficult to directly assess lens deformability.

An ideal method for evaluating the mechanical behaviour of the eye lens would need to keep the lens intact and in contact with its surrounding structures, such as zonular fibres and the ciliary muscle, so that the measuring process can best mimic the in vivo lens deformation. Such stretching devices have been developed in the past [27,36,37,38,39]. In these stretching devices, four to eight jaws were designed to be fixed to the ciliary muscle [27,36] or isolated anterior eye segment using either glue or pins [27,36,37,38]. However, most of these stretching devices used independent arms to control the stretching jaws [27,36,39,40,41], which means they cannot precisely ensure equal movements of all the jaws nor equal stretching of the eye sample in all directions. Recently, Webb et al. [42] developed a stretcher with a plate of an eight-line sliding chute that can drive the movements of all eight arms at the same time. However, in a design that has a line-shaped sliding chute, the speed of centrifugal movement of the stretching arm is not linearly related to the speed of rotation of the stretcher, and as a result the stretching speed is difficult to control. These stretchers have been used for analysing the accommodative response of the lens [42,43], as well as the functions of zonular fibres [40,41].

In the present study, an automatically controlled stretcher to be used for mechanical assessment of the anterior eye segment is described. In this design, the plate with a sliding chute that drives the movement of stretching arms was developed according to the spiral of Archimedes, so that a rotation of the stretcher at a constant speed can produce a likewise constant speed of centrifugal movement of the stretching arm. Inspired by the mechanical design of surgical forceps, a clamping mechanism is proposed that can more effectively hold the stretched sample to avoid soft tissue slippage. Moreover, the stretcher is integrated with a high-precision torque transducer for real-time measurement, and all units are controlled by a compact single-chip so that all stretching experiments can be conducted automatically. In the following sections, the design and the theoretical analysis of the proposed device are explained. Calibration and spring-based validation experiments to evaluate its main functionality are then described. Finally, a feasibility study with porcine eyes is presented.

## 2. Materials and Methods

### 2.1. Mechanical Design

#### 2.1.1. Stretcher

The eye stretcher consists of a base, eight loading units, and a circular actuating plate, as shown in Figure 1a. The base is installed on the gear system and has eight rectangular grooves and a cylindrical cavity to constrain the motion of the loading units and actuating plate. The loading units are placed in eight rectangular grooves which are evenly distributed circumferentially, pointing to the centre of the cavity. The actuating plate is installed on the loading units and constrained by the internal surface of the cylindrical cavity, which guarantees the coaxial alignment of the actuating plate and the cavity. Six stop pins are placed on the top to restrain the movement of the plate in the vertical direction. Eight identical curved grooves, that are evenly distributed in the actuating plate, are positioned based on the spiral of Archimedes according to the polar coordinate equation:(1)r=a+bθ
where *r* and *θ* indicate polar radius and polar angle; *a* = 35 mm and *b* = 20/π mm/rad are two geometric constants. Differentiating Equation (1) with respect to time, a velocity mapping between the radius and angle can be generated as
(2)$$\dot{r}=b\dot{\theta }$$
which indicates that the point on the spiral performs a uniform concentric or eccentric motion as long as $\dot{\theta }$ is constant. A cylindrical pin is fixed in each of the eight loading units (Figure 1b) so that the units can move in the spiral groove simultaneously (Figure 1c). Combining the constraints of the rectangular and spiral groove, the loading unit is able to stretch the eye sample at a constant strain rate when the actuating plate rotates uniformly. Two outward bars of the plate are designed to receive the driving torque from the motor, which is transmitted through the gear system described below.

The loading unit is composed of a base, a jaw and a slider, as shown in Figure 1b. The pin that is designed to move in the spiral groove is fixed on the slider of the loading unit to drive its movement. The slider is hollow and can move linearly along the base of the loading unit. The front end of the base is equipped with a jaw to hold the eye sample tightly. At the rear of the jaw and the front of the slider, the same oblique angle is devised to strengthen the clamping force when the slider is pushed forward. The rear parts of the slider and the cuboid have paired clamping teeth to maintain the position of the slider, ensuring that they can be uncoupled by pushing the rear of the slider downward.

#### 2.1.2. Gear System

The gear system is composed of a motor, a torque sensor and two sets of gears (Figure 2). This system is used to transmit the driving torque from the motor to the stretcher. The modules of the gear for set 1 and set 2 are 0.8 and 0.9, respectively. The number of teeth for the first set of gears that is designed to connect the motor and the sensor are *n*_1_ = 40 and *n*_2_ = 80, respectively. The number of teeth of the second set of gears that connect the sensor and stretcher are *n*_3_ = 100 and *n*_4_ = 100, respectively. The transmission ratio *i* could be calculated as
(3)$$i=i_{1}\cdot i_{2}=\frac{n_{1}}{n_{2}}\cdot \frac{n_{3}}{n_{4}}=0.5$$
where *i*_1_ and *i*_2_ indicate the ratio of the first and second gear set, respectively. The torque sensor measures the driving torque, and the influence of friction can be eliminated by the additional calibration method described below. The prototype of the stretcher and gear system were fabricated by 3-dimensional (3D) printing using Somos EvoLVe 128 resin.

#### 2.1.3. Stretching Force Sensing

As the polar coordinate equation is defined in Equation (1), the Cartesian coordinates (Figure 2) of the spiral can be derived as
(4)$$\begin{array}{l} x=r\cos\theta =\left(a+b\theta \right)\cos\theta \\ y=r\sin\theta =\left(a+b\theta \right)\sin\theta \end{array}$$
where the right end of a spiral is defined as the origin (*θ* = 0). Differentiating Equation (4), the unit tangent vector of the spiral at the point *θ = θ*_0_ can be obtained from
(5)$${k_{t}|}_{\theta =\theta_{0}}=\frac{{\left[f_{x}^{\prime }\left(\theta_{0}\right),f_{y}^{\prime }\left(\theta_{0}\right),0\right]}^{T}}{{\left\|{\left[f_{x}^{\prime }\left(\theta_{0}\right),f_{y}^{\prime }\left(\theta_{0}\right),0\right]}^{T}\right\|}_{2}}\;\mathrm{where}\;f_{x}^{\prime }\left(\theta \right)=\frac{dx}{d\theta }=b\cos\theta -\left(a+b\theta \right)\sin\theta \;f_{y}^{\prime }\left(\theta \right)=\frac{dy}{d\theta }=b\sin\theta +\left(a+b\theta \right)\cos\theta$$

The unit normal vector, along which the groove of the actuating plate exerts force ***F****_n_* on the pin of the loading unit, can then be calculated as
(6)$$k_{n}=k_{t}\times {\left[0,0,1\right]}^{T}$$

With respect to the point ***r***, the reacting force of ***F****_n_* can generate the moment ***τ***
(7)$$\tau =r\times f_{n}k_{n}$$
where *f_n_* is the scalar of ***F****_n_*. Since the eight spiral grooves are the same and ***F****_n_* is applied in the *xoy*-plane, the total torque needed to drive the loading units is
(8)$$T=8\tau_{z}$$

*T* can be measured by the sensor, and *f_n_* can then be calculated by Equation (7) and the stretching force *f_r_*, the magnitude of the vector ***F****_r_*, derived from
(9)$$f_{r}=\frac{f_{n}}{\left\|r\right\|}k_{n}^{T}r$$

### 2.2. Electronics and Control

The proposed system uses an all-in-one servo motor with a built-in driver and encoder for actuation. The brushless DC servo can be switched between three modes: open-loop, speed or position control. This is regulated and can be further programmed using the embedded chip of the driver (32-bit MCU, 72 MHz). The torque sensor for measurement is based on a bridge strain gauge, the measurement range of which is within −0.3 to 0.3 N.m (WTN-56, WEABU Electronics Technology Co., Ltd., Dongguan, China). A custom-designed circuit board (70 mm × 68 mm) is used in addition to off-the-shelf components. The board includes a main control chip, a power regulating module, a voltage amplifier module, a CAN transceiver module and a communication module. The main control chip is based on the STM32F407 series (Cortex™-M4 core, 168 MHz, STMicroelectronics, Plan-les-Ouates, Switzerland), and the communication module is based on a TTL-WIFI communication module (DT-06, Shenzhen Doctors of Intelligence & Technology Co., Ltd., Shenzhen, China). Based on the above electronic control architecture, a Python-based Graphical User Interface (GUI, written by python 3.8.8) was developed to control the motion of the stretcher system and to record the collected torque. Both serial communication and WIFI-based teleoperation can be used in the above system. In terms of motion regulation, the system allows for speed control over a certain range. The operating software allows the stretcher to move to a pre-set fixed angle, which is achieved through the basic PID control of the motor. In addition, the system also allows master–slave operation under manual control via a gamepad.

### 2.3. Experimental Measurement

#### 2.3.1. Calibration of the Resistance Force

Measurement of the resistance was conducted by controlling the actuator to rotate the stretcher without any loaded sample. During the measurement, the stretcher was controlled in angle mode, which included predefined parameters through the GUI, such as the angle of rotation and the speed of rotation. The angle of rotation was set to 40 degrees and the speed of rotation was set to 2, 5 and 10 degrees per second (dps), respectively. For each speed of rotation, the measurement was repeated three times. In each measurement, the stretcher was actuated for two trips, namely the forward movement and the backward movement. The torque measured by the force sensor at corresponding angles during the forward movement was recorded for further analysis. The experimentally measured resistance force data were firstly filtered by the Savitzky–Golay (savgol) filter with a window size of 11, after which data measured at same rotational speed were averaged and finally curve-fitted using the linear regression method. The window size was determined by practical tests under the consideration that no distortions were caused to the overall trend of the filtered data. The curved-fitted results can be used for resistance deduction in subsequent experiments with the spring and the porcine eye samples.

#### 2.3.2. Validation Experiment Based on Spring Stretching

To verify the proposed device, a spring with a known elastic constant was chosen for the stretching experiment. During the experiment, the spring was suspended between two jaws in opposite directions (Figure 3a) and stretched with the actuation of the motor and mechanism. Given that the formula of the Archimedes spiral is known, the angle data and torque data have a defined relationship, which could be converted to the force and displacement (F–D) relationship using Equations (4) and (9). Based on the measured F–D curves, we can analyse the measured spring constant and compare it with the true spring constant. The F–D data were filtered by the 2^nd^-order ButterWorth filter using a cutoff frequency of 2 Hz to filter out signals with spectrum characteristics.

#### 2.3.3. Feasibility Test Using Porcine Eyes

Three fresh porcine eyes were used for the validation of the stretcher. The porcine eyeballs were provided by local abattoirs and used within eight hours of death. The cornea of the eyeball and the posterior half of the eyeball were removed, leaving the lens, its surrounding zonular fibres, ciliary process and a ring of sclera. The sclera was then carefully cut into eight segments (Figure 3b) to be clamped by the eight jaws of the loading unit of the stretcher (Figure 3c). To load the sample, the stretcher was maintained at the zero-degree of rotation position and the eight jaws remained open. The anterior eye segment was carefully placed at the centre of the stretcher using tweezers, which can extend each scleral piece while closing the corresponding jaw. The loaded sample can then be extended under the actuation of the motor. The F–D data measured with porcine eye samples were also filtered using the 2nd-order ButterWorth filter of 2Hz cutoff frequency.

## 3. Results

Resistance calibration tests were performed with the stretcher operated at three different speeds, 2 dps, 5 dps and 10 dps when no sample was loaded. The torque measured by the sensor and its relationship to the relative rotation angle of the stretcher at three speeds are plotted in Figure 4. At each speed, the torque angle data were recorded three times, and repeatability was demonstrated at all three speeds (mean torque between 15 to 40 degrees of rotation angle: 0.0584 ± 0.2619 × 10^−5^ Nm at the speed of 2 dps, 0.0587 ± 0.7645 × 10^−5^ Nm at the speed of 5 dps and 0.0576 ± 0.0547 × 10^−5^ Nm at the speed of 10 dps). At 10 dps there were fewer recorded data (Figure 4c) than at 2 dps and 5 dps (Figure 4a,b). This is because the sensor was set to record the data at a constant time interval and because less time was taken for the stretcher to cover the same angle range when running at a higher speed. The torque values show similar magnitude and trends with increasing angles among three different speeds; therefore, 5 dps was selected for later validation experiment with a spring and feasibility test using porcine eyes.

The three repeated torque rotation angle data points at 5 dps were firstly averaged, and then the data in the region of 5 to 40 degrees were selected and fitted with a linear regression line to be used for resistance deduction in later experiments (Figure 5). The reason for selecting this region is that it is the effective working region for the stretcher when it is loaded with the spring and porcine eyes. The linear regression line has a relatively low slope of 0.0002 Nm/deg, suggesting that the resistance is quite small and relatively constant with changing rotation angles. The root-mean-square error (RMSE) of the fitting was found to be 0.0019 Nm, accounting for 3.1834% of the maximum resistance value.

The validation test was performed by loading a spring of a known spring constant to the stretcher while repeating the measurement three times. The recorded torque data were firstly calibrated by deducting the resistance shown in Figure 5 from the measured data. The calibrated torques were then converted to stretching force values according to Equations (4)–(9) and the rotation angle was converted to distance in radial direction according to Equations (1) and (2). The resultant F–D curves of the spring were firstly filtered and then fitted using linear regression, as shown in Figure 6. The calculated stiffness for the three repeated measurements were found to be 0.2690 N/mm, 0.2984 N/mm and 0.2847 N/mm, respectively, in comparison to the theoretical value of 0.2800 N/mm.

Three porcine eye segments, including the lens, the zonular fibres, the ciliary muscle and the sclera (equally cut into eight segments as described in Section 2, were prepared and carefully loaded to the stretcher for the testing of feasibility. Images showing different degrees of stretching for one eye are displayed in Figure 7. At the start of the stretching process, the eight loading units are located close to each other (Figure 7a), and the zonular fibres are relaxed so the sensor cannot detect effective torque values. As the stretcher starts to rotate and the loading units move in distal directions (Figure 7b–d), the sensor starts to report detected values when the zonular fibres are taut, causing the lens to deform. Figure 7e shows the maximal level of stretching.

Effective readings of the measured torque with corresponding angles of rotation were used to convert the F–D curves during the lens deformation (Figure 8). In general, the force that deforms the porcine lens is within the range of 200 to 450 mN, and for all three porcine samples the force increases with stretching. The F–D curve of each sample was fitted with a linear regression. Individual variations in the lens stiffness can be seen: the first porcine lens has a higher stiffness than the other two, as it shows the steepest slope of the regression line; the second porcine lens demonstrates a lower regression slope (Figure 8b), therefore lower stiffness. The successful completion of the porcine eye test indicated the effectiveness of the experimental setup, particularly in meeting the required measurement range.

## 4. Discussion

This study describes the design, development and testing of a new automatically controlled multi-axis stretching device for the mechanical evaluation of soft tissues and, more specifically, of the anterior eye segment. The developed stretching device has the following unique features that have not been demonstrated in any previous design: (1) the stretcher can rotate at a constant speed in a centrifugal movement with a sliding chute that propels the stretching arms’ motion, in accordance with Archimedes’ spiral; (2) the device incorporates a clamping mechanism that can guarantee a tight gripping of the soft tissue inspired by the design of surgical forceps; (3) the device has been automated with the integration of the motor, torque sensor and transmission mechanism. These features make the current design theoretically superior to most existing similar devices available commercially or reported in the literature [27,36,37,38,39,40,41,42,43,44] and have potential applications for testing a wider range of isotropic soft tissues and materials.

The measurements of the porcine eye samples reported in this study verified the effectiveness of the device, and more specifically indicated whether the stretching could be achieved and measurement data recorded. Biological samples such as porcine eye lenses are quite pliable, and therefore the required stretching forces need to be low. Experimental results demonstrate that the device is capable of detecting such low-level forces after resistance correction, which is very important for the validation of the device. Unlike the validation experiment using springs, the porcine eye sample does not have a theoretical stiffness value with which it can be compared. Results in Figure 8 show variations in force values and the range of displacements that were attributed to variations in size and stiffness among different biological samples. Previous measurements from porcine, monkey and human eyes have reported force–displacement data during ex vivo stretching [37,38,40,41,43,44]. Due to the different forms of data presentation and the lack of stretching force values for the porcine eye, these previous results are not directly comparable. The best evidence that can be used for comparison is from Cortés et al. [37], showing force–displacement curves for human eye samples aged 42 years and above which are of the same order of magnitude as those identified in the present study.

The main application of the device is for quasi-static measurements of the specimens at low speeds, so the core of the calibration is dependent on its internal resistance. According to the corresponding experimental measurements, the resistance characteristics show good repeatability under the same conditions and agreement among experiments using different speed settings. Linear regression of the resistance resulted in a relatively low slope, suggesting that the values are approximately constant at different angles. The resistance is largely from the friction generated by the linear slider and other connecting elements during low-speed movements. The focus of subsequent studies will be the analysis of the sources that cause resistance and further reduce the resistance by improving the design and optimising the materials of device components.

The accuracy of the measurement of this experimental device is validated by the spring test with repeated measurements. Linear regression of the force–displacement curve gives the measured stiffness value of the spring, which is well matched with the true spring constant value. The porcine eye lenses, which are easy to harvest from local abattoirs, demonstrate a similar magnitude of elasticity as that of humans [12,31] and are therefore an ideal animal model for experimental usage [45,46]. Studies have shown that eye lenses of other species, i.e., the rabbit and mouse [47,48,49,50], have different stiffnesses from human lenses, which may be related to their specific visual needs for living. It is important to investigate the dynamic mechanical behaviour of eye lenses of animals living in different conditions to understand deeply how the lens mechanics influence the accommodation and presbyopia.

The simulation of accommodation experimentally needs to take into account all physiological aspects of the process in order to be as representative as possible of the in vivo situation. In the healthy eye, the lens adjusts its shape within fractions of a second and the shape change depends on object position. A shortcoming of this stretcher is that the sensor is set for a constant time interval. Hence, measurements are undertaken at different speeds and do not always produce the same number of data points for analysis and for comparison with the biological in situ lens. Filtered curve fits to experimental measurements on porcine samples show only general trends and do not take into account the fluctuations in the experimental data, particularly for samples shown in Figure 8a,c. These fluctuations may have arisen because the lenses were removed from the eye and had undergone some dehydration. In future work, investigations will be focused on improving the mechanical design by incorporating various standard parts to compare their respective performance and enhancing sensor resolution, as well as on validating this stretching device on many more fresh eye samples from a wider range of experimental animals.

## Figures and Tables

**Figure 1 bioengineering-10-00142-f001:**
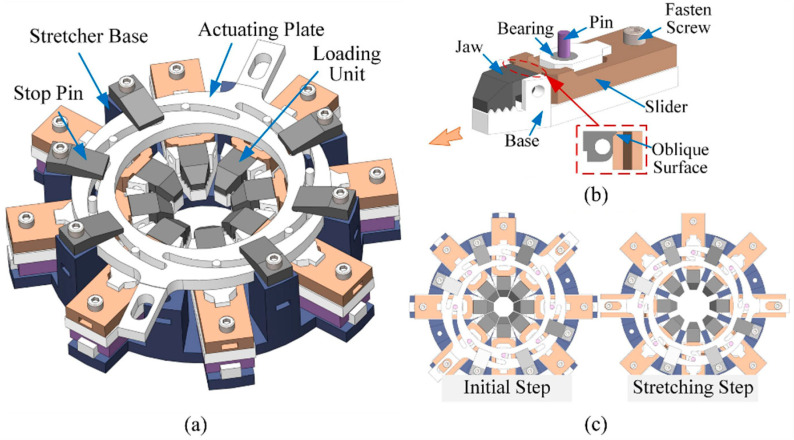
Schematic representation of the stretcher’s mechanical structure: (**a**) whole stretcher; (**b**) loading unit; (**c**) top view of stretched and unstretched positions.

**Figure 2 bioengineering-10-00142-f002:**
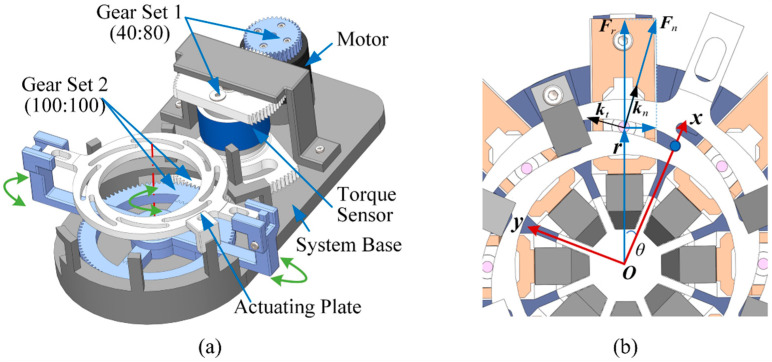
Schematic representation to illustrate (**a**) the torque transmission and the measuring unit and (**b**) the relationship between the measured torque and the stretching force.

**Figure 3 bioengineering-10-00142-f003:**
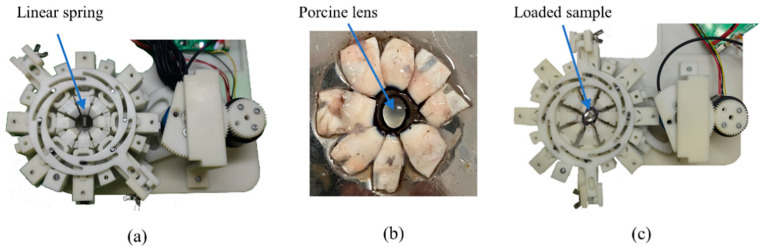
Experimental setup for (**a**) the spring-based validation test; (**b**) the prepared anterior porcine eye sample and (**c**) the porcine-eye-based feasibility test using the proposed stretcher.

**Figure 4 bioengineering-10-00142-f004:**
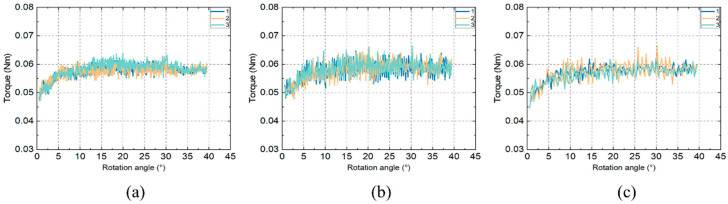
The relationship between the rotation angle and the measured torque for the speed at (**a**) 2 dps; (**b**) 5 dps and (**c**) 10 dps during the calibration experiment.

**Figure 5 bioengineering-10-00142-f005:**
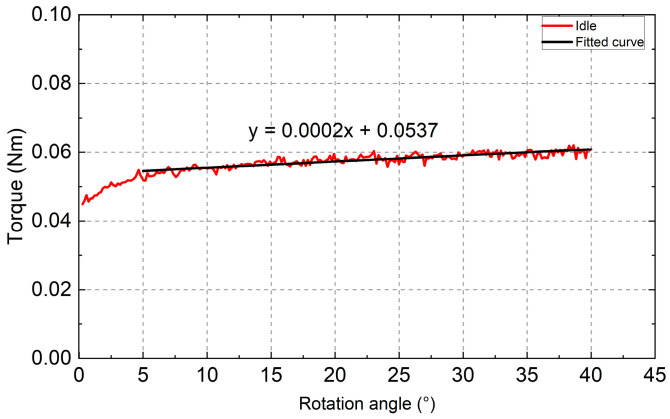
Fitting result of the selected resistance curve with the fitting parameters shown.

**Figure 6 bioengineering-10-00142-f006:**
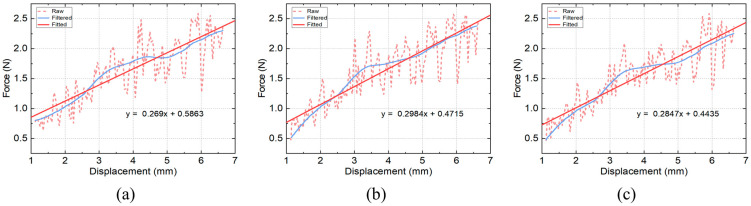
Measurement and fitting results of the stretching experiment of the spring for three repeated times. Force displacement curves of spring validation test obtained from (**a**) the first measurement, (**b**) the second measurement and (**c**) the third measurement.

**Figure 7 bioengineering-10-00142-f007:**
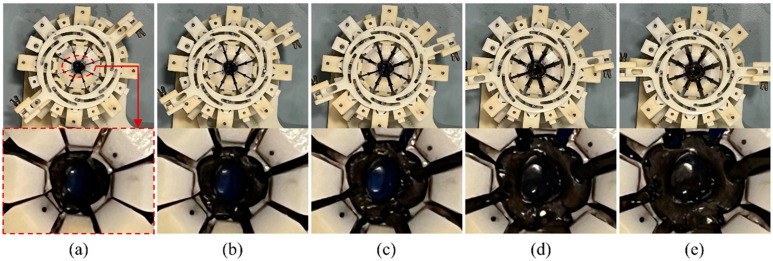
Images showing different degree levels of stretching for an anterior porcine eye sample. (**a**) the eight loading units are close to each other at the start of the stretching, the eight loading units move in distal directions with (**b**) showing zonular fibres in relaxed state, (**c**) showing zonular fibres in tensional state and (**d**) shown expansion of lens equatorial size, (**e**) the eight loading units at the maximal level of stretching.

**Figure 8 bioengineering-10-00142-f008:**
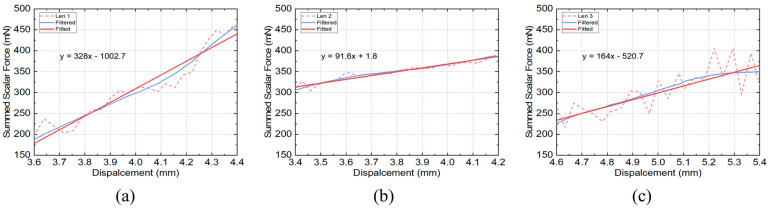
Measured force–displacement data and their corresponding filtered and fitted results of three different porcine eye samples. (**a**) porcine eye sample one, (**b**) porcine eye sample two and (**c**) porcine eye sample three.

## Data Availability

Not applicable.

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
