# Peer review of "Design of an Automatically Controlled Multi-Axis Stretching Device for Mechanical Evaluations of the Anterior Eye Segment"

_bioengineering, 2023, doi:10.3390/bioengineering10020142_

Round 1
Reviewer 1 Report
The authors present a novel prototype to measure zonular stretch or lens capsule stretch via an automatically controlled stretcher: the plate with a sliding chute that drives the movement of stretching arms with a rotation of the stretcher at a constant speed producing a comparable constant speed of centrifugal movement of the stretching arm. The device is well tested in vitro. The paper is well written. Of course additional work is required to validate its usefulness in experimental live animals
Reviewer 2 Report
This paper presents a 3D-printed stretching device prototype that integrates a mechanical stretcher, motor, torque sensor and data transmission mechanism system.
The idea looks creative, and the design of the mechanical parts looks OK; however, there is a lot of work that needs to be done on the signal processing side.
Digital noise (high-frequency signal) can be seen clearly in figures 4, 5, 6 & 8; therefore, it is not recommended to fit such data without filtering the noise effectively, which is not the case in the current presentation.
According to the text, the experimentally measured data was averaged, then filtered by the Savitzky-Golay (savgol) filter and finally curve fitted using the linear regression method. It does not look appropriate to average the data before applying the filter, as noise must have been averaged with the signal before going through the filter. Typically, no need to average the data at all if the torque signal will be filtered anyway.
The averaging technique is not mentioned; is it a moving average? If so, what was the sliding window size?
Figure 6’s displacement is not showing the range from 0 mm to 1 mm, which is the most critical physiological range when testing biological tissue. Still, noise to signal ratio is very high, at least visually. Why not display it when validating against a linear spring?
Reviewer 3 Report
The authors present an original and interesting system for multi-axial testing of tissues. The manuscript is well written and some results on a few samples are presented.
Major
I have one major concern on the application of the system to non-homogeneous and in particular to anisotropic tissues, as most soft biological tissues are.
From what I have understood reading the paper, the 8 displacements are equal as imposed by the motor and as consequence of the Archimede spiral equations. Nevertheless I believe that the resistance force for each one of the 8 direction may vary depending on the possible variation of the sample mechanical properties with direction of loading. Since you put only one force sensor, the measured value will be the average of the resistance forces in the different directions. If I am not right, please better explain in the methods section how you calculate each of the eight forces by using one single sensor. If I am right, please underline that the system is valid only for isotropic tissues. Is the lens tissue of the eye an isotropic material ? Or can it be considered mostly isotropic ?
Minor
Equation 2, r needs a dot
Line 216, how is the sample carefully entered ?
Line 227 can you change the data acquisition frequency by programming so to make it adequate for higher velocities ?
Figure 6 and 8: for what previously discussed, the force should be indicated by the average force along the different loading directions
Line 291 after revision please consider to modify the sentence "have potential applications for testing of wider range of soft tissues and materials" to "have potential ....wider range of isotropic tissues and materials"
Round 2
Reviewer 2 Report
Please justify the Savitzky-Golay filter's window size of 11
Please justify the choice of 2nd order ButterWorth filter and the cut-off frequency of
2Hz
This reviewer disagrees with the author's response; figure 6's displacement is still not showing the range from 0 mm to 1 mm in subplots (a) and (b); please add this range in a revised version. Original measured signal and filtered signal need to be displayed over this range.
Figure 8 showed an initial force measurement even when the displacement was zero. Also, the magnitude of this force varied between subplots. Why is there a force at zero displacements? Perhaps friction of mechanical components? Why is it not the same? Please discuss. Also, consider rewriting the figure caption to describe sub figures contents better.
Add a paragraph toward the end of the discussion about the study's limitations, including sensors' resolution, filtering and curve fittings.
Reviewer 3 Report
Authors have cleared my doubts and corrected as requested
Author Response
Thanks for your comments.